# Emerging Roles of Ubiquitination in Biomolecular Condensates

**DOI:** 10.3390/cells12182329

**Published:** 2023-09-21

**Authors:** Peigang Liang, Jiaqi Zhang, Bo Wang

**Affiliations:** 1State Key Laboratory of Cellular Stress Biology, School of Life Sciences, Faculty of Medicine and Life Sciences, Xiamen University, Xiamen 361102, China; 18750288411@163.com (P.L.); zhangjq641@nenu.edu.cn (J.Z.); 2Shenzhen Research Institute of Xiamen University, Shenzhen 518057, China

**Keywords:** ubiquitin, biomolecular condensates, liquid–liquid phase separation, stress granules, autophagy

## Abstract

Biomolecular condensates are dynamic non-membrane-bound macromolecular high-order assemblies that participate in a growing list of cellular processes, such as transcription, the cell cycle, etc. Disturbed dynamics of biomolecular condensates are associated with many diseases, including cancer and neurodegeneration. Extensive efforts have been devoted to uncovering the molecular and biochemical grammar governing the dynamics of biomolecular condensates and establishing the critical roles of protein posttranslational modifications (PTMs) in this process. Here, we summarize the regulatory roles of ubiquitination (a major form of cellular PTM) in the dynamics of biomolecular condensates. We propose that these regulatory mechanisms can be harnessed to combat many diseases.

## 1. Introduction

Eukaryotic cells have evolved prominent compartments to fulfill more efficient and sophisticated regulation of biochemical reactions and signaling. A considerable proportion of the cellular compartments are not membrane-bound; these are now collectively known as biomolecular condensates. The thermodynamics of biomolecular condensates is fundamentally distinct from those of membrane-bound compartments, such as mitochondria. For instance, the dynamics of biomolecular condensates is significantly faster than those of membrane-bound organelles, allowing biomolecular condensates to rapidly respond to acute environmental or intracellular stimuli [1]. Biomolecular condensates are implicated in an ever-growing list of biological pathways, such as transcription, translation, etc. [2]. Aberrant dynamics of biomolecular condensates is deemed an important contributor to many diseases, such as cancer and neurodegeneration [2,3].

Work from the last decade has established the widely appreciated role of liquid–liquid phase separation (LLPS) in the dynamics of biomolecular condensates. LLPS refers to the spontaneous demixing of biomolecular solutions, including protein and/or nucleic acid, into a condensed liquid phase that is biophysically and biochemically distinct from the surroundings [4]. Often, protein LLPS is dependent on unstructured regions—also known as intrinsically disordered regions (IDRs)—that engage proteins in either weak intra- or inter-molecular promiscuous interactions. Alternatively, protein interaction motifs facilitate LLPS and provide multivalency to membrane-less compartments by stoichiometrically interacting with binding partners [5]. These two mechanisms, however, are not mutually exclusive. Instead, both can cooperatively govern biomolecular LLPS and, therefore, the dynamics of complex membrane-less networks in a context-dependent fashion. While the mechanisms of how a protein’s primary amino-acid sequence dictates the material properties of biomolecular condensates are understood to a great extent [6], how posttranslational modifications (PTMs) in proteins, such as phosphorylation and ubiquitination, function as a molecular switch to fine-tune the dynamics of biomolecular condensates is less well understood.

Ubiquitination is a widespread and reversible PTM that covalently attaches ubiquitin, a small conserved regulatory protein (76 residues), to the target proteins (most commonly lysine residues). It involves stepwise catalyzation by a cascade of enzymes that are categorized as E1, E2, and E3. This modification involves either a single ubiquitin protein (mono-ubiquitylation) or a chain of ubiquitin (poly-ubiquitylation and multi-ubiquitination). In the case of poly-ubiquitination, the secondary ubiquitin molecules are always attached to one of the seven lysine residues (known as K6, K11, K27, K29, K33, K48, and K63 ubiquitination) or the N-terminal methionine (known as M1 ubiquitination) of the previous ubiquitin molecule.

K48- and K63-linked poly-ubiquitin chains represent the two most prevalent types of ubiquitin chains in cells [7,8]. K48-linked poly-ubiquitin serves as a crucial modification that mediates protein degradation [9]. A disruption in cellular redox balance leads to the extensive accumulation of oxidized proteins. K48-linked ubiquitination plays an essential role in the degradation pathway of these oxidized proteins [10]. In the event of DNA damage, ring finger protein 8 (RNF8), an E3 ubiquitin–protein ligase, applies K48-ubiquitin modifications to key regulatory factors at the damage sites. These modifications are subsequently removed, leading to a rearrangement of the signaling complex that facilitates the proper assembly of downstream factors at the damage site and, ultimately, driving DNA repair [11]. K63 ubiquitination is of significant importance as an initial event in signal transduction pathways associated with both innate and adaptive immunity [12]. K63-linked ubiquitination plays a central role in regulating NF-κB activation and inflammatory responses [13]. Moreover, K63-linked chains are known to promote autophagy within cells. When K63-linked ubiquitination is enhanced, the resulting ubiquitin-positive inclusions exhibit a higher degree of colocalization with the autophagy receptor p62 [14].

Various forms of non-canonical protein ubiquitination (such as K6, K11, and K27) exist in cells, and they play pivotal roles in the degradation of obsolete proteins across diverse signaling pathways [15]. K6-linked ubiquitination, for instance, is integral to mitophagy. When mitochondria become dysfunctional, PTEN-induced kinase 1 (PINK1) targets the mitochondrial outer membrane and phosphorylates ubiquitin associated with the outer membrane proteins [16]. This action recruits Parkin, an E3 ubiquitin ligase, which subsequently decorates damaged mitochondrial outer membrane proteins with K6- and K63-linked chains, thereby designating the mitochondria for autophagy [17]. In addition, K11- and K27-linked ubiquitination has been reported to control the cell cycle, DNA damage responses, and viral infections, among other cellular processes [18,19,20,21].

In principle, protein ubiquitination can affect biomolecular condensates through two distinct mechanisms. It either alters the molecular configuration of protein constituents and, therefore, the physiochemical properties of biomolecular condensates or provides valency for binding partners that harbor ubiquitin-binding domains (UBDs) or ubiquitin-associated domains (UBAs) within the molecular network. Ubiquitin accumulates in patients with neurodegenerative diseases, such as amyotrophic lateral sclerosis (ALS); these likely involve disrupted autophagy activities and the homeostasis of stress granules. Therefore, ubiquitination is implicated in governing the dynamics of these disease-related biomolecular condensates. Indeed, evidence supporting this hypothesis started to emerge over the years. Here, we summarize this evidence and highlight the key role of ubiquitination in regulating stress-granule dynamics and autophagy initiation. Based on these pre-clinical studies, we postulate that targeting biomolecular condensates through ubiquitination holds great promise for combatting diseases such as ALS.

## 2. Ubiquitin in Stress Granule Dynamics

Stress granules are membrane-less organelles located in the cytoplasm, formed in response to various environmental threats, such as high temperature, oxidative stress, and viral infections [22]. Stress granules are messenger ribonucleoprotein complexes (mRNPs) containing stalled mRNA, RNA binding proteins (RBPs), translation initiation factors, and other proteins [23]. RBPs, such as Ras GTPase-activating protein (SH3 domain)-binding proteins 1 and 2 (G3BP1/2), interact with untranslated mRNAs and undergo LLPS, giving rise to stable “cores” [24]. These cores then recruit additional stress granule nucleators, resulting in the formation of more dynamic peripheral “shell”-like structures [24]. Once the external stimulus subsides, stress granules disassemble to release the RBPs and mRNAs, which coincides with translation recovery [25]. This cellular mechanism enables the efficient recycling of stress-granule components, including proteins and mRNAs, to avoid de novo synthesis [25]. The dynamics of stress granules is extensively regulated by PTMs, such as phosphorylation and methylation [26]. Recent studies have also highlighted the roles of ubiquitination in stress-granule dynamics, including assembly, disassembly, and degradation.

Various elements of ubiquitination-associated machinery, including E1 ubiquitin-activating enzyme, E3 ubiquitin ligases (the histone E3 ligase 2 (Hel2), anaphase promoting complex (APC), tripartite motif protein family members 21 (TRIM21), etc.), deubiquitylases (ubiquitin-specific-processing proteases USP5, USP10, and USP13), ubiquitin-binding proteins (the ubiquitin-binding proteins ubiquilin 2 (UBQLN2), histone deacetylase 6 (HDAC6)), the 26S proteasome, and valosin-containing protein (VCP/p97), have been found in the SG proteome [24,27,28,29,30,31,32,33,34,35]. Furthermore, two independent studies have demonstrated that stress granules induced with different stressors contain free ubiquitin species not attached to any target proteins [28,36].

Acute inhibition of protein ubiquitination by the E1 inhibitor, TAK243, does not appear to prevent stress granule formation in response to heat shock or arsenite [37,38], suggesting that active protein ubiquitination is likely not essential for stress granule formation, per se. On the contrary, several ubiquitin-binding proteins have been directly linked to stress-granule formation, including UBQLN2, which acts as a proteasome shuttle to deliver the ubiquitinated substrates for degradation [39]. UBQLN2 forms liquid condensates both in vitro and in vivo, which are inhibited with mono-ubiquitin or poly-ubiquitin chains [31]. UBQLN2 localizes to stress granules and interact with several stress granule proteins, including fused in sarcoma (FUS) [40,41]. By fluidizing the FUS-RNA complex, UBQLN2 negatively regulates stress granule assembly [40]. Therefore, ubiquitination may serve as a switch between UBQLN2 recruitment to stress granules and UBQLN2-dependent shuttling of ubiquitinated stress-granule components. Moreover, HDAC6, as a unique member of the class II deacetylase, containing a C-terminal zinc finger domain with a high binding affinity to free ubiquitin as well as mono- and poly-ubiquitinated proteins, is critical for stress-granule formation [30,42]. Pharmacological inhibition or genetic ablation of HDAC6 disrupts stress-granule assembly [30]. The seemingly conflicting findings from these studies likely stem from the context-dependent regulation of ubiquitination in stress-granule formation.

In addition to these ubiquitin-binding proteins, deubiquitinating enzymes (DUBs), such as USP5 and USP13, have also been implicated in stress-granule formation. Recent studies on heat-induced stress granules containing K48- and K63-linked ubiquitin chains demonstrate that the depletion of USP5 or USP13 accelerates stress-granule assembly, while stress-granule disassembly, upon returning cells to normal temperature, is significantly repressed [28]. USP5 is suggested to regulate heat-induced stress granules by hydrolyzing unanchored ubiquitin chains, whereas USP13 controls stress granules by deubiquitylating protein-conjugated ubiquitin chains [28]. Furthermore, the deletion of Hel2 is shown to increase stress-granule formation as well [35], and the inhibition of the E3 ubiquitin ligase APC and its regulatory subunit Cadherin-1 (Cdh1) in primary cortical neurons also leads to increased stress-granule formation [33]. These findings collectively highlight the critical role of the ubiquitin system in regulating stress-granule assembly in response to various cellular stressors.

Different stressors elicit distinct ubiquitination patterns within the stress-granule proteome. Specifically, heat shock results in prominent ubiquitination of the stress-granule constituents, whereas arsenite, another common inducer of stress granules, does not [37]. This heat-shock-induced ubiquitination plays a crucial role in the resumption of cellular activities and stress-granule disassembly [37]. One of the stress-granule scaffold proteins, G3BP1, is ubiquitinated under heat-shock conditions. The ubiquitination of G3BP1 weakens the stress-granule-specific interaction network, and in the meantime, enhances its interaction with the endoplasmic-reticulum-associated protein and engages the ubiquitin-dependent segregase VCP/p97. As a segregase, VCP/p97 efficiently extracts the ubiquitinated G3BP1 from the stress-granule network and, ultimately, leading to stress-granule disassembly [43]. Notably, G3BP1 is recently demonstrated to be ubiquitinated by TRIM21 in the arsenite-induced stress granules [34]. Ubiquitinated G3BP1 exhibits a higher threshold concentration required for LLPS, which is probably accountable for the compromised stress-granule assembly. Additionally, it is reported that heat stress leads to the ubiquitination of various stress-granule components, including FUS, TAR DNA-binding protein 43 (TDP-43), and certain DEAD box proteins [37,44], which may function through similar mechanisms as summarized above to disentangle stress granules.

The fate of stress granules and the mechanism of their elimination are context-dependent. The ubiquitination of stress-granule components not only impacts stress-granule assembly/disassembly but also elimination under different conditions. It is demonstrated that stress granules induced by short-term heat stress (30 or 60-min) can be dissolved via disassembly, whereas stress granules formed under prolonged 90-min heat stress require autophagy for clearance [43]. In both cases, the segregase VCP is essential for stress-granule dissolution, either through direct disassembly or autophagy-dependent degradation, which may involve differential regulation of VCP via cofactors that are capable of directly binding to ubiquitin [32,43]. While the VCP-FAF2 (FAS-associated factor 2) complex is relatively well characterized in stress-granule disassembly, the molecular mechanisms underlying the VCP-mediated degradation of stress granules through autophagy are yet to be fully uncovered (Figure 1).

The role of ubiquitination in stress granules is directly linked to human diseases, particularly neurodegenerative disorders. The dysregulation of stress-granule dynamics has been implicated in pathological conditions such as ALS, frontotemporal dementia (FTD), and Huntington’s disease (HD) [45,46,47]. Notably, the cytoplasmic inclusions observed in ALS are frequently subject to ubiquitination [48,49]. TDP-43, a major component of the ubiquitinated inclusions in most ALS patients, is reported as a target for ubiquitination [48]. Additionally, as the disease progresses, misfolded superoxide dismutase 1 (SOD1) proteins form cytoplasmic ubiquitinated inclusions that aggravate over time [49]. Similarly, ubiquitin is also enriched in the cytotoxic protein aggregates in Alzheimer’s patients. The Alzheimer’s disease-related protein Tau undergoes complex coacervation through electrostatic interactions with nucleic acids, and aberrant Tau LLPS might contribute to the pathogenesis of Alzheimer’s disease [50]. Interestingly, a recent report shows that the mono-ubiquitination of Tau weakens its condensation with polyanions, such as RNA and heparin [51]. Nevertheless, whether this specific Tau ubiquitination/deubiquitination modification occurs in vivo, and whether it directly contributes to the Tau aggregates observed in patient tissues awaits further investigation [51]. Together, these findings highlight the critical role of ubiquitination in governing stress-granule dynamics and its implications for various neurodegenerative diseases. Further research into the mechanisms underlying stress-granule ubiquitination may provide new insights into the pathogenesis of neurodegenerative diseases and offer potential therapeutic targets for these disorders.

## 3. Ubiquitin in Autophagy

Autophagy, an evolutionarily conserved process mediated by lysosomes, encapsulates a segment of cytoplasmic constituents by creating a double-membrane autophagosome, which is then targeted to the lysosome for degradation and subsequently macromolecule recycling [52,53]. Serving as a vital source of materials and energy for cells under stress, autophagy selectively eliminates the misfolded or surplus proteins to maintain cellular equilibrium [53,54]. Protein ubiquitination plays a pivotal role in this process. The initiation and nucleation phases of autophagosome formation are typically governed by ubiquitination [55]. In addition, ubiquitin-mediated regulation also oversees the later stages of autophagosome formation and maturation [56]. Furthermore, under prolonged stress conditions, ubiquitin-mediated protein turnover is employed as the primary mechanism for terminating autophagy to prevent the detrimental effects of excessive autophagy flux [57,58].

Recent research has revealed that protein LLPS plays a crucial role in the assembly of autophagosomes and the condensation of autophagy substrates [59,60,61]. In budding yeast, the regulation of the cytoplasm-to-vacuole targeting (Cvt) pathway, an autophagy-related pathway trafficking proteolytic enzymes into vacuoles, is influenced by condensates formed through Ape1 phase separation. Ape1 utilizes phase separation to create a pliable assembly structure capable of engaging with the endoplasmic reticulum. This interaction is made possible by the strategic high-density positioning of Atg19 on its surface. Such interaction culminates in the sequestration of the Ape1 condensate into the Cvt vesicle [62]. The formation of a pre-autophagosomal structure (PAS), a dynamic and transient structure that regulates autophagosome formation on the vacuole, involves interactions between Atg1 and several proteins, including Atg13, Atg17, Atg29, and Atg31 [63]. When dephosphorylated, Atg13 forms an Atg1 complex with Atg1 and Atg17. This cross-linking activity prompts the Atg1 complex to form liquid-like condensates in vitro, which is responsible for organizing the PAS in vivo [64]. Currently, little is known regarding the mammalian Atg1 homologue/ULK1 complex undergoing phase separation to regulate autophagy initiation [65].

Under conditions of proteotoxic stress, such as the inactivation of chaperone proteins or proteasomes, proteins tagged with ubiquitin rapidly conglomerate into aggregates [66]. The creation of these aggregates is largely reliant on p62, as its depletion results in ubiquitinated proteins being more scattered within cells [67]. p62, as one of the first selective autophagy receptors discovered in mammals, is believed to mediate the selective autophagic degradation of poly-ubiquitinated protein aggregates [68]. p62 consists of a self-association PB1 domain, a UBA domain, and a LC3/Atg8 interaction region [69]. In vitro experiments indicate that p62 protein alone cannot undergo phase separation, even at very high concentrations. The phase separation of p62 only occurs when it is mixed with long ubiquitin chains that are unattached to proteins. The formation of p62 condensates is influenced by various factors, including the valency of ubiquitin chains, and its binding affinity towards ubiquitin chains [70]. Paradoxically, there is evidence that K63 poly-ubiquitin chains depolymerize p62 filaments [71]. This might be caused by the excessive valency conferred by the poly-ubiquitin chains, which therefore favors heterotypic ubiquitin-p62 interactions and disfavors homotypic p62 oligomerization. Therefore, these multi-ubiquitin chains not only earmark proteins for autophagy degradation but also serve as the activation signals to initiate cargo condensation (Figure 2). Proteins within protein aggregates are typically misfolded and inactive, making biochemical reactions within the aggregates unlikely. Conversely, proteins within droplets retain their conformation and activity and can diffuse within the droplet and its surrounding environment. The distinction between protein aggregates and phase-separated droplets may have significant functional implications for the selective autophagy and other aspects [67,70,72].

Under physiological conditions, the ubiquitination-mediated phase separation of the autophagic cargoes is regulated by multiple components [73]. The collaboration of p62, NBR1, and TAX1BP1 cargo receptors plays a crucial role in the formation and autophagic degradation of ubiquitin-enriched condensates [74,75]. Specifically, the phase separation of p62 and ubiquitin chains is considered as the primary driving force for the formation of autophagic cargo condensates [70,72]. NBR1 directly boosts condensation by offering a high-affinity UBA domain and simultaneously recruits TAX1BP1 and FIP200 to the p62-ubiquitin condensates. Moreover, the NBR1 UBA domain has a stronger affinity for ubiquitin compared to the UBA domain of p62, suggesting that hetero-oligomeric NBR1–p62 complexes may have a stronger affinity for ubiquitinated substrates than homo-oligomeric p62 [76]. Mitophagy is a specialized type of selective autophagy that eliminates damaged mitochondria. During mitophagy, mitochondrial proteins are often poly-ubiquitinated, which is recognized by p62 and several other receptors such as OPTN, NBR1, etc. [77]. The function and regulation of p62 in mitophagy can vary with the upstream inducers of mitophagy. In the context of celastrol-induced mitophagy, p62 cooperates with Nur77 to promote the efficient clearance of mitochondria. Nur77, a nuclear receptor, is ubiquitinated and translocated to the damaged mitochondria due to celastrol treatment. Ubiquitinated mitochondrial Nur77 interacts with the UBA domain of p62, and together they may form membrane-less condensates which can isolate damaged mitochondria. Additional interactions between Nur77 IDR and p62 domains liquidize the Nur77–p62 condensates, which is essential for tethering mitochondria to the autophagic machinery [78]. In summary, the selection and condensation of ubiquitin chain-dependent autophagic cargos regulated by phase separation rely on the joint action of multiple adaptor proteins.

Various proteins, such as TDP-43 and Tau, whose mutations are linked to ALS/FTD have the ability to form droplets that can evolve into gels or aggregates under pathological conditions [79,80]. The failure to promptly eliminate these aggregated proteins culminates in the build-up of ubiquitin-positive inclusions that are cytotoxic [50,81]. In addition, defects in the ubiquitin-related autophagy machinery can also trigger disease pathogenesis. For instance, point mutations in p62 have been discovered in both Paget’s disease of bone (PDB) and ALS [82,83]. Notably, 25–50% of all familial PDB patients harbor mutations in p62, predominantly located in the UBA domain [84,85]. Among them, M404T and G411S mutants significantly impair p62 phase separation with multi-ubiquitin chains. Furthermore, when p62 M404T and G411S mutations are incorporated into *p62*^-/-^ cells, both hinder the formation and autophagic degradation of p62 bodies [70]. TBK1 is found to phosphorylate the S403 site within the UBA domain of p62. In ALS-FTD, TBK1 mutations result in a reduced p62 phosphorylation, which diminishes the ability of p62 to bind to ubiquitin chains. This decreases the capacity of p62 to form condensates, and, ultimately, leads to the build-up of toxic proteins [70,86]. These studies suggest that protein phase separation mediated by ubiquitination is a prerequisite for the normal function of autophagy under physiopathological conditions and is vital for maintaining cellular homeostasis.

## 4. Ubiquitin in Other Biomolecular Condensates

In addition to the relatively well-defined roles of ubiquitination in autophagosome formation and stress-granule dynamics, ubiquitination has also been implicated in several other biomolecular condensates.

Nuclear speckles are typical membrane-less biomolecular condensates located in the interchromatin-space of the nucleus, serving as the reservoir for RNA processing and splicing factors for mRNA alternative splicing [87]. Nuclear speckle assembly is driven by two large scaffold proteins, SRRM2 and SON [88,89]. Direct evidence that nuclear speckles arise through LLPS is lacking, which is at least partially due to technical challenges. Nevertheless, the existing evidence supports an active role of ubiquitination in nuclear speckle dynamics, likely through modulating LLPS. Several ubiquitin-associated proteins are detected in the nuclear speckles. Specifically, speckle-type POZ protein (SPOP) is a substrate adaptor of the cullin3-RING ubiquitin ligase that is frequently mutated in various solid tumors [90]. SPOP localizes to nuclear speckles, which is reliant on high-order oligomerization and substrate binding [91,92]. Cancer-related SPOP mutations disrupt its oligomerization, LLPS, nuclear speckle localization, and ubiquitin signaling [91]. The biological consequences of these SPOP mutants on nuclear speckle dynamics have yet to be explored. USP42 is a deubiquitylase bearing a positively-charged C-terminus that is sufficient for USP42 LLPS and nuclear speckle formation [93]. Importantly, the deubiquitination activity of USP42 is critical for its nuclear speckle localization and the partitioning of specific clients, such as PLRG1, into nuclear speckles [93]. While these studies implicate an interesting role of ubiquitin in nuclear speckle dynamics, the detailed biochemical mechanisms of how SPOP, USP42, or other ubiquitin-associated factors impact LLPS and nuclear speckle dynamics remain unclear and require further investigation.

Cells assemble dynamic and transient biomolecular condensates in response to external stress. An interesting type of nuclear condensate that contains proteasomal constituents and poly-ubiquitinated proteins is recently demonstrated to be triggered by acute hyperosmotic stress [94]. The formation of such nuclear proteasomal condensates is driven by the ubiquitin-binding shuttle protein RAD23B. The binding of RAD23B to poly-ubiquitin chains via its two UBA domains promotes RAD23B condensation. Subsequently, RAD23B recruits various proteasome-associated proteins, including the E3 ligase E6-AP and VCP, into these nuclear condensates via its ubiquitin-like (UBL) domain. RAD23B condensates are proteolytic centers that actively degrade specific ribosomal proteins. Although many important questions regarding RAD23B condensates remain unanswered, this study exemplifies the essential role of ubiquitin in novel biomolecular condensates. In budding yeast, nuclear proteasomes translocate to the cytoplasm and are sequestered in motile and reversible biomolecular condensates, called proteasome storage granules (PSGs), during quiescence [95]. PSG formation requires free ubiquitin, although how exactly ubiquitin functions in this process is enigmatic [96]. PSGs, which sequester proteasomes during quiescence and protect them from autophagic degradation, rapidly mobilize upon re-entry into the cell cycle [97]. The freed proteasomes from PSG dissolution likely participate in removing poly-ubiquitinated proteins from other proteostasis quality-control mechanisms to preserve their dynamic reversibility and adaptability [98]. Whether PSGs are present in mammalian cells or not remains an intriguing question.

Ubiquitination has also been shown to play a key role in biomolecular condensates involved in signaling transduction. The nuclear factor NF-κB pathway governs a variety of cellular processes, including immune responses, etc. [99]. The key effector of the NF-κB pathway is a family of structurally-related transcription factors, including NF-κB1/p50, NF-κB2/p52, RelA/p65, RelB, and c-Rel [99]. NF-κB proteins are normally sequestered in the cytoplasm by inhibitory proteins such as IκB family members. The suppressive effect of IκB on NF-κB is relieved when IκB is phosphorylated by the IκB kinase complex (IKK), which is composed of IKKα, IKKβ, and a regulatory subunit named NF-κB essential modulator (NEMO) or IKKγ. A recent investigation has demonstrated that the poly-ubiquitin chain activates IKK and therefore NF-κB signaling through NEMO LLPS [100,101,102]. NEMO phase separation is dependent on the presence of a specific poly-ubiquitin chain (K63 or M1). Depending on the upstream stimuli, several E3 ligases including TRAF6, HOIP, cIAP1, or TRAF2 are detected within the NEMO condensates, in a context-dependent fashion. Notably, IKK is activated within the NEMO condensates [63]. Conversely, the overexpression of two DUBs, CYLD and A20, which are well-known inhibitors of NF-κB, significantly reduces NEMO condensation and inhibits IKK activation [100]. In an independent study, similar findings that NEMO activates the NF-κB signaling via ubiquitin-dependent phase separation have been reported [101]. These studies together highlight the context-dependent regulation of NEMO condensates fulfilled through ubiquitination, among other mechanisms. DLV2 is a pivotal mediator of Wnt signaling that dictates cell-fate determination during development [103]. A recent study shows that DLV2 undergoes ubiquitination-dependent phase separation [104]. Interrogating the function of WWP2, an E3 ligase responsible for DVL2 ubiquitination or DVL2 ubiquitination by direct mutagenesis leads to disrupted DVL2 phase separation. More importantly, phase-separated DVL2 condensates activate Wnt signaling by sequestering the components of negative regulators, known as the destruction complex [104]. Together, the ubiquitinated proteins and possibly ubiquitin moieties can directly participate in signaling pathways through modulating phase separation.

## 5. Summary and Future Perspectives

In this review, we summarize the current knowledge regarding the functions of ubiquitination in biomolecular condensates (Table 1). As more efforts are being devoted to this surging field, we expect an expanding list of ubiquitination-coupled biomolecular condensates. Notably, the detailed regulatory mechanisms of how ubiquitination regulates biomolecular condensates remain largely unrevealed. Specifically, how ubiquitination alters the local structure and chemistry near the modification sites to impact either inter- or intra-molecular interactions and, subsequently, the molecular thermodynamics within biomolecular condensates has been inadequately uncovered. The identity of enzymes fulfilling ubiquitination in these processes is often enigmatic. Moreover, do deubiquitylating enzymes participate in the reverse processes? The effect of ubiquitination on the physiochemical properties of molecular condensates is poorly characterized. The pathophysiological significance of ubiquitination-coupled biomolecular condensates is almost unexplored. Answering these fundamental questions can be challenging and entails multidisciplinary approaches but it is of great interest given that ubiquitin is prominently accumulated in foci that are thought to originate from the disturbed metabolism of biomolecular condensates in patients with diseases such as ALS.

Indeed, targeting the ubiquitin pathway holds great promise for treating diseases, especially neurodegeneration. Numerous preclinical studies, taking either genetic and/or pharmacological approaches, have demonstrated the feasibility of targeting this pathway to mitigate or, in some rare cases, even reverse disease phenotypes. The best-known example is HDAC6 [105,106]. The consequences of targeting HDAC6 can vary, however. Both inhibiting and enhancing the activity of HDAC6 have been shown to circumvent disease progression in different preclinical animal models [107,108,109,110]. The seemingly conflicting observations may be due to the context-dependent regulation of HDAC6 in different diseases. Similarly, p62 has also emerged as a highly promising candidate to combat neurodegeneration [111]. Mounting evidence suggests that p62 is neuroprotective, probably through modulating autophagy [112,113]. Although compounds directly acting on p62 are currently lacking, small molecules stimulating p62-mediated autophagy facilitate the selective elimination of disease-related protein aggregates; some of these compounds are in clinical trials for treating neurodegeneration [111]. Additionally, given the well-documented functions of USPs in inhibiting stress-granule formation [28,29,114] and the frequent dysregulation of USPs in neurodegeneration [115], boosting the activities of USPs might therefore hamper aberrant stress-granule accumulation and subsequent protein aggregation. It is worth mentioning that ubiquitin-related proteins are multifunctional. Therefore, a deeper understanding of the specific functions of these elements of ubiquitin-associated machinery in biomolecular condensates is required to achieve better therapeutic effects. The best mechanism by which to selectively modulate biomolecular condensate-associated activities to minimize the potential side effects will remain a major challenge for therapeutic considerations.

**Table 1 cells-12-02329-t001:** Ubiquitination-dependent regulation of biomolecular condensates.

Biomolecular Condensates	Ubiquitin	Effect on Phase Separation	Reference
Arsenite, or heat-induced stress granules	Poly-ubiquitin	Promotes LLPS	[30]
Heat-induced stress granules	Mono-ubiquitin	Causes disassembly	[28]
Arsenite, or heat-induced stress granules	K63 poly-ubiquitin	Causes disassembly	[34,43]
p62 condensates	Poly-ubiquitin	Promotes LLPS	[70]
p62 condensates	Mono-ubiquitin	Causes disassembly	[69]
Proteasome condensate	Mono-ubiquitin	Causes disassembly	[116]
UBQLN2 phase separation	Poly-ubiquitin	Causes disassembly	[31]
Dvl2 phase separation	Poly-ubiquitin	Promotes LLPS	[104]
NEMO phase separation	Poly-ubiquitin	Promotes LLPS	[100]

## Figures and Tables

**Figure 1 cells-12-02329-f001:**
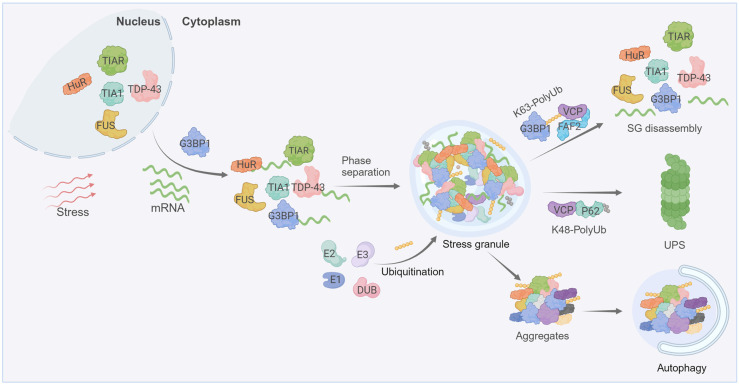
Stress-granule dynamics are tightly regulated by ubiquitination. Cells under stress inhibit the translation initiation of mRNAs, leading to stress granules’ formation that involves the phase separation of RBPs (such as G3BP1, T-cell intracellular antigen 1 (TIA-1), ELAV-like protein 1 (HuR), TIA1-related protein (TIAR) etc.) and mRNAs. Both the assembly and disassembly of stress granules are regulated by PTMs, including ubiquitination. The disassembly of stress granules is enhanced through the K63-linked ubiquitination of G3BP1, subsequently fostering the interaction between ubiquitin chains and VCP. Mutations in RBPs or prolonged stress cause the transition of stress granules into pathological aggregates. Ubiquitination governs the clearance of these aggregates, with K48-linked ubiquitin chains directing degradation through the ubiquitin-proteasome system (UPS), whereas K63-linked ubiquitination is associated with the autophagic degradation pathways. Created with BioRender.com.

**Figure 2 cells-12-02329-f002:**
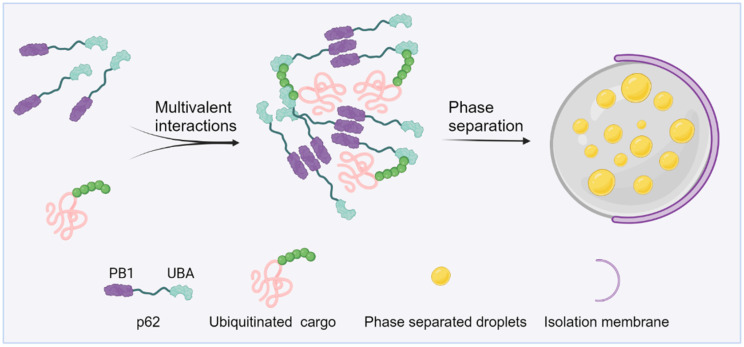
The process of autophagic cargo segregation is driven by poly-ubiquitin chain-induced p62 phase separation. p62 protein forms oligomers via the PB1 domain and also binds to ubiquitin through the UBA domain. Upon reaching the threshold concentration required for LLPS, the formation of p62 condensates occurs. Subsequently, other client proteins are recruited to these p62 condensates that further mature and ultimately are broken down through autophagy. Created with BioRender.com.

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
