# Peer review of "Emerging Roles of Ubiquitination in Biomolecular Condensates"

_cells, 2023, doi:10.3390/cells12182329_

Round 1

Reviewer 1 Report

In the review article by Liang et al. the authors discuss the role of ubiquitination in different biomolecular condensates, mainly stress granules and p62 bodies in autophagy, as well as nuclear speckles and other condensates involved in signal transduction.

The last years there is an increased interest and number of studies concerning liquid-liquid phase separation and the formation of biomolecular condensates within cells. These are interesting cellular mechanisms that play a crucial role in a large number of biological pathways enabling the cell to react fast to different environmental stimuli and accomplish specific tasks i.e. biochemical reactions or signal transduction. The authors give a nice overview and explain the role of ubiquitination in the formation and dynamics of these condensates. Moreover, they attempt to connect it with a number of neurodegenerative diseases.

I find the review well written, nicely summarizing the findings on the role of ubiquitin in different biomolecular condensates.

1.    It is thought that ubiquitinated cargo recognition occurs through p62 filaments formed by PB1 domain polymerization. A sequence of at least 4 ubiquitins crosslinks p62 filaments forming the condensates. On the other hand, there is evidence that free ubiquitin depolymerizes p62 filaments. How can these two opposing effects of the same molecule be rationalized in the context of cargo recognition and isolation?  

2.    Since ubiquitination of specific proteins/enzymes affect the assembly/disassembly of biomolecular condensates that could lead to pathological conditions, would targeting these proteins offer some kind of treatment for these neurodegenerative diseases? For example the deubiquitinating enzymes in the stress granules or p62 in the case of selective autophagy.

3.    Could ubiquitin itself be used as a carrier to introduce drugs in specific targets?

4.    Could the authors think of ways to make use of this knowledge to treat ALS and the other mentioned neurodegenerative disorders?

Author Response

  •  It is thought that ubiquitinated cargo recognition occurs through p62 filaments formed by PB1 domain polymerization. A sequence of at least 4 ubiquitins crosslinks p62 filaments forming the condensates. On the other hand, there is evidence that free ubiquitin depolymerizes p62 filaments. How can these two opposing effects of the same molecule be rationalized in the context of cargo recognition and isolation?  

We thank the reviewer’s critical question. The condensation formation of p62 is dependent on many factors. The most important force driving p62 phase separation is likely the molecular stoichiometries of p62 and ubiquitin within the aqueous solution. In addition, the valency of ubiquitin chains also alters the p62 phase separation. When excessive poly-ubiquitin chains are present in the reaction system, heterotypic ubiquitin-p62 interactions are favored and homotypic p62 oligomerization are disfavored, which therefore dissolves p62 filament formation. Many similar findings are reported previous. For example, low concentration RNA facilitates phase separation of SARS-CoV-2 N protein whereas excessive RNA inhibits its phase separation (PMID: 33479198). We have now added a sentence in the revised manuscript to reflect this concept.

Paradoxically, there is evidence that K63 poly-ubiquitin chains depolymerizes p62 filaments [57]. This might be caused by the excessive valency conferred by the poly-ubiquitin chains, which therefore favors heterotypic ubiquitin-p62 interactions and disfavor homotypic p62 oligomerization.

  • Since ubiquitination of specific proteins/enzymes affect the assembly/disassembly of biomolecular condensates that could lead to pathological conditions, would targeting these proteins offer some kind of treatment for these neurodegenerative diseases? For example, the deubiquitinating enzymes in the stress granules or p62 in the case of selective autophagy.
  • Could ubiquitin itself be used as a carrier to introduce drugs in specific targets?
  • Could the authors think of ways to make use of this knowledge to treat ALS and the other mentioned neurodegenerative disorders?

We appreciate the reviewer’s constructive comments. We believe the #2-4 fall into the same category: whether the ubiquitin-associated biomolecular condensates can be targeted for therapeutic purposes. The same issue was raised by another reviewer as well. We have therefore elaborated on this in the “Summary and Future perspectives” section.

Indeed, targeting the ubiquitin pathway holds great promises for treating diseases, especially neurodegeneration. Numerous preclinical studies, taking either genetic and/or pharmacological approaches, have demonstrated the feasibility of targeting this pathway to mitigate or, in some rare cases, even reverse disease phenotype. The most prominent example is HDAC6 [90,91]. The consequence of targeting HDAC6 can vary, however. Both inhibiting and enhancing HDAC6 activity have been shown to circumvent disease progression in different preclinical animal models [92-95]. The seemingly conflicting observations may be due to the context-dependent regulation of HDAC6 in different diseases. Similarly, p62 has also emerged as a highly promising candidate to combat neurodegeneration [96]. Mounting evidence suggest that p62 is neuroprotective, probably through modulating autophagy [97,98]. Although compounds directly acting on p62 are currently lacking, small molecules stimulating p62-mediated autophagy facilitate selective elimination of disease-related protein aggregates, some of which are in clinical trials for treating neurodegeneration [96]. Additionally, given the well-documented functions of USPs in inhibiting stress granule formation [14,15,99] and the frequent dysregulation of USPs in the neurodegeneration [100], boosting the activities of USPs might therefore hamper aberrant stress granule accumulation and subsequent protein aggregation. It is worth mentioning that the ubiquitin-related proteins are multifunctional. Therefore, a deeper understanding of the specific functions of these ubiquitin-associated machinery, whether or not biomolecular condensates are involved, is required to achieve better therapeutic effects. How to selectively modulate the biomolecular condensate-associated activities to minimize potential side effects will remain a major challenge for therapeutic considerations.

Reviewer 2 Report

Ubiquitination could potentially influence the formation and dissolution of biomolecular condensates. Ubiquitin-mediated modifications on specific proteins within condensates might regulate their interactions and phase separation behavior. Ubiquitination can serve as a molecular tag for specific proteins, targeting them for recognition by ubiquitin-binding domains (UBDs) present in other proteins. This targeting mechanism could potentially facilitate the selective recruitment of certain ubiquitinated proteins into biomolecular condensates, influencing their composition and function. Ubiquitination is closely linked to protein degradation, and proteins within biomolecular condensates might undergo ubiquitin-dependent degradation, allowing for the turnover of components and maintaining the dynamic nature of the condensates.

Biomolecular condensates often form in response to cellular stress. A previous study demonstrated that in budding yeast, upon entry into a quiescence state, proteasome subunits re-localize from the nucleus to the cytoplasm. Furthermore, these cytoplasmic proteasome reservoirs, known as proteasome storage granules (PSGs) rapidly mobilizes upon exit from quiescence (PMID: 21402786). Earlier study demonstrated that human shuttling protein UBQLN2 undergoes liquid-liquid phase separation (LLPS) under physiological conditions and colocalizes with stress-granules (SGs), whereby interaction with Ub or ubiquitinated substrates reverses LLPS and may shuttle clients out of SGs (PMID: 29526694). Moreover, a study indicated the role of UBA and UBL domain in LLPS of UBQLN2 (PMID: 34029402). Another important study in the field showed that stress-induced ubiquitination is dispensable for the formation of SGs and shutdown of cellular pathways. Furthermore, ubiquitination is crucial for disassembly of SGs and for resumption of cellular activities upon recovery from stress (PMID: 34739326). All the above mentioned works evidently demonstrated a definitive functional correlation between biomolecular condensation and Ubiquitin-proteasome system. Importantly, the authors in this manuscript illustrated specific aspect of ubiquitination in biomolecular condensation.

However, there are certain aspect, authors should consider including here:

1) In the introduction section, line 59-61.

“Only poly-ubiquitylations on defined lysine, K48 and K29, confer degradation signals by the proteasome, while other poly-ubiquitylations (e.g., on K63, K11, and K6) are implicated in an array of cellular activities such as endocytic trafficking, inflammation, etc.”

Author must elaborate on this. The degradation signal presented by diverse ubiquitin topology (Following articles are good example for this e.g. PMID: 29378950, PMID: 27746020, PMID: 28165462).

2) In figure1, authors should mark the mRNA, as it will add more clarity for the readers.

3) A Table on “different biomolecular condensates and involvement of Ubiquitin” with references could be an excellent addition.

4) In section 2 (Ubiquitin in stress granule dynamics), author must consider writing about PSGs and role of ubiquitin in this (PMID: 28768827).

5) The review lack some valuable therapeutic aspects of studying biomolecular condensation. Authors might consider adding this in the future perspective section.

Author Response

1) In the introduction section, line 59-61.

“Only poly-ubiquitylations on defined lysine, K48 and K29, confer degradation signals by the proteasome, while other poly-ubiquitylations (e.g., on K63, K11, and K6) are implicated in an array of cellular activities such as endocytic trafficking, inflammation, etc.”

Author must elaborate on this. The degradation signal presented by diverse ubiquitin topology (Following articles are good example for this e.g. PMID: 29378950, PMID: 27746020, PMID: 28165462).

We thank the reviewer’s suggestion. We have now described the function and regulation of different ubiquitin chain modification. Please refer to line 60-84 in the revise manuscript.

2) In figure1, authors should mark the mRNA, as it will add more clarity for the readers.

We marked the mRNA in the Figure 1 as suggested.

3) A Table on “different biomolecular condensates and involvement of Ubiquitin” with references could be an excellent addition.

We thank the reviewer’s suggestion. In the “Summary and Future perspectives” section, we have now made a table summarizing the different roles of ubiquitin modifications in the biomolecular condensates. Please see Table 1.

Biomolecular condensates

Ubiquitin

Effect on Phase Separation

Reference

Arsenite, or heat-induced stress granules

Poly-ubiquitin

Promotes LLPS

[16]

Heat-induced stress granules

Mono-ubiquitin

Causes disassembly

[14]

Arsenite, or heat-induced stress granules

K63 poly-ubiquitin

Causes disassembly

[20,29]

p62 condensates

Poly-ubiquitin

Promotes LLPS

[56]

p62 condensates

Mono-ubiquitin

Causes Disassembly

[55]

Proteasome condensate

Mono-ubiquitin

Causes Disassembly

[101]

UBQLN2 phase separation

Poly-ubiquitin

Causes Disassembly

[17]

Dvl2 phase separation

Poly-ubiquitin

Promotes LLPS

[89]

NEMO phase separation

Poly-ubiquitin

Promotes LLPS

[85]

4) In section 2 (Ubiquitin in stress granule dynamics), author must consider writing about PSGs and role of ubiquitin in this (PMID: 28768827).

We agree with the reviewer that PSGs could indeed be an important part of our article. We hope the reviewer would agree that the description about PSGs might be a better fit for the section 4 “Ubiquitin in other biomolecular condensates”. We have now summarized the key points about PSGs in the section 4.

In budding yeast, nuclear proteasomes translocate into the cytoplasm and are sequestered in motile and reversible biomolecular condensates called proteasome storage granules (PSGs) during quiescence [80]. PSG formation requires free ubiquitin, although how exactly ubiquitin functions in this process is enigmatic [81]. PSGs, which sequester proteasomes during quiescence, protecting them from autophagic degradation, rapidly mobilizes upon re-entry into cell cycle [82]. The freed proteasomes after PSG dissolution likely participate in removing polyubiquitinated proteins from other proteostasis quality control mechanisms, preserving their dynamic reversibility and adaptability [83]. Whether PSGs are present in the mammalian cells or not remains an intriguing question.

5) The review lacks some valuable therapeutic aspects of studying biomolecular condensation. Authors might consider adding this in the future perspective section.

We acknowledge the reviewer for this valuable comments. The same issue was raised by another reviewer. We have therefore elaborated on this in the “Summary and Future perspectives” section.

Indeed, targeting the ubiquitin pathway holds great promises for treating diseases, especially neurodegeneration. Numerous preclinical studies, taking either genetic and/or pharmacological approaches, have demonstrated the feasibility of targeting this pathway to mitigate or, in some rare cases, even reverse disease phenotype. The most prominent example is HDAC6 [90,91]. The consequence of targeting HDAC6 can vary, however. Both inhibiting and enhancing HDAC6 activity have been shown to circumvent disease progression in different preclinical animal models [92-95]. The seemingly conflicting observations may be due to the context-dependent regulation of HDAC6 in different diseases. Similarly, p62 has also emerged as a highly promising candidate to combat neurodegeneration [96]. Mounting evidence suggest that p62 is neuroprotective, probably through modulating autophagy [97,98]. Although compounds directly acting on p62 are currently lacking, small molecules stimulating p62-mediated autophagy facilitate selective elimination of disease-related protein aggregates, some of which are in clinical trials for treating neurodegeneration [96]. Additionally, given the well-documented functions of USPs in inhibiting stress granule formation [14,15,99] and the frequent dysregulation of USPs in the neurodegeneration [100], boosting the activities of USPs might therefore hamper aberrant stress granule accumulation and subsequent protein aggregation. It is worth mentioning that the ubiquitin-related proteins are multifunctional. Therefore, a deeper understanding of the specific functions of these ubiquitin-associated machinery, whether or not biomolecular condensates are involved, is required to achieve better therapeutic effects. How to selectively modulate the biomolecular condensate-associated activities to minimize potential side effects will remain a major challenge for therapeutic considerations.

Reviewer 3 Report

Over the last decades, we realized that besides the classical membrane-surrounded subcellular compartments, within the cells can also be found membraneless subcellular compartments. They can be found in the cytoplasm but also in membrane-surrounded compartments, such as the nucleus. They are dynamic structures whose occurence relies on liquid-liquid phase separation. They partecipates in a wide spectrum of biological processes, hence when malfunctioning they contribute to pathological conditions. Being dynamic structures their assembly and disassembly are tightly controlled by different stimuli and post-translational modification (PTMs), including phosphorylation and ubiquitination.  In the manuscript titled "Emerging Roles of Ubiquitination in the Biomolecular Condensates" Liang P. and colleagues review the emerging roles played by ubiquitination in finely regulating liquid-liquid phase separation and thus the occurrence of such membraneless compartments. Ubiquitination is carried out through stepwise reactions involving E1, E2, and E3 enzymes. E3 ubiquitin ligases are responsible for selecting the different substrates. Ubiquitination "codes" are then read by different ubiquitin binding proteins and ubiquitinated substrates might be delivered to proteasome where degraded or Ub-chains might act as docking site and contribute to intrecellular signaling. The manuscript presents some interesting insights, however prior to publication a couple of amendments are required. The legend in Figure 1 has to be implemented by detailing what are the different molecules. For example, the green ribbons I guess are mRNAs. However, currently, it is not specified. Similarly what do HuR, TIA, and so on, stand for? Acronyms have to be detailed when cited for the first time. Furthermore, in lines 150-152 the authors state "While the VCP-FAF2 complex is relatively well characterized in stress granule disassembly, the molecular mechanisms underlying VCP-mediated degradation of stress granule through autophagy are yet to be fully uncovered (Figure 1)". Thus it seems that VCP-FAF2 has a well-recognized role in stress-granule disassembling, hence it would be valuable to depict it in Figure 1. Line 55: It seems that multiubiquitination has been omitted, why? Line 92: to my knowledge, HDAC6 is the only family member harboring a Zinc finger domain thus enabling its interaction with Ub, while all the other family members do not "handle" Ub. The issue should be clarified. Please add a reference line 169 "...are frequently subject to ubiquitination (REF)." Few typos for example lines 166-167 "...stress granule dynamics and has been implicated in...".

English language is quite fine, minor editing is required. Few typos are scattered throughout the main text.

Author Response

  1. The legend in Figure 1 has to be implemented by detailing what are the different molecules. For example, the green ribbons I guess are mRNAs. However, currently, it is not specified. Similarly, what do HuR, TIA, and so on, stand for?

We appreciate the reviewer for pointing this issue out-similar issues were raised by another reviewer. We have now clearly labeled mRNAs in the figure and specify the different molecules in the figure legend.

  1. Acronyms have to be detailed when cited for the first time. Furthermore, in lines 150-152 the authors state "While the VCP-FAF2 complex is relatively well characterized in stress granule disassembly, the molecular mechanisms underlying VCP-mediated degradation of stress granule through autophagy are yet to be fully uncovered (Figure 1)". Thus, it seems that VCP-FAF2 has a well-recognized role in stress-granule disassembling, hence it would be valuable to depict it in Figure 1. Line 55: It seems that multiubiquitination has been omitted, why? Line 92: to my knowledge, HDAC6 is the only family member harboring a Zinc finger domain thus enabling its interaction with Ub, while all the other family members do not "handle" Ub. The issue should be clarified.

We appreciate the reviewer’s suggestions. A: We detailed the acronyms as needed. B: VCP-FAF2 was depicted in the Figure 1 of our original submission, but the font was not large enough. We have now enlarged the font to increase the readability. C: We added the multi-ubiquitination. D: We clarified the fact that HDAC6 is the only class II deacetylase that have a zinc finger domain enabling binding to ubiquitin. “Moreover, HDAC6, as a ubique member of the class II deacetylase containing a C-terminal zinc finger domain with high binding affinity to free ubiquitin as well as mono- and poly-ubiquitinated proteins, is critical for stress granule formation [16,28].

  1. Please add a reference line 169 "...are frequently subject to ubiquitination (REF)." Few typos for example lines 166-167 "...stress granule dynamics and has been implicated in...".

We corrected these errors as requested, thanks!